# ACE2 Knockout Mice Are Resistant to High-Fat Diet-Induced Obesity in an Age-Dependent Manner

**DOI:** 10.3390/ijms25179515

**Published:** 2024-09-01

**Authors:** Valéria Nunes-Souza, Natalia Alenina, Fatimunnisa Qadri, Valentina Mosienko, Robson Augusto Souza Santos, Michael Bader, Luiza Antas Rabelo

**Affiliations:** 1Max Delbrück Center for Molecular Medicine in the Helmholtz Association, 13125 Berlin, Germany; valeria.nsouza@ufpe.br (V.N.-S.); fqadri@mdc-berlin.de (F.Q.); valentina.mosienko@bristol.ac.uk (V.M.); mbader@mdc-berlin.de (M.B.); 2Department of Physiology and Pharmacology, Federal University of Pernambuco, Recife 50670-901, Brazil; 3National Institute of Science and Technology in Nanobiopharmaceutics (Nanobiofar), Belo Horizonte 31270-901, Brazil; robsonsant@gmail.com; 4School of Physiology, Pharmacology & Neuroscience, University of Bristol, Bristol BS8 1TD, UK; 5Department of Physiology and Biophysics, Federal University of Minas Gerais, Belo Horizonte 31270-901, Brazil; 6DZHK (German Center for Cardiovascular Research), Partner Site Berlin, 10785 Berlin, Germany; 7Charité Universitätsmedizin Berlin, Corporate Member of Freie Universität Berlin, 10117 Berlin, Germany; 8Institute for Biology, University of Lübeck, 23562 Lübeck, Germany; 9Laboratory of Cardiovascular Reactivity, Metabolic Syndrome Center, Institute of Biological Sciences and Health, Federal University of Alagoas, Maceió 57072-900, Brazil

**Keywords:** angiotensin-converting enzyme type 2, obesity resistance, glycemic signaling, aging

## Abstract

Angiotensin converting enzyme 2 (ACE2) presents pleiotropic actions. It hydrolyzes angiotensin I (AngI) and angiotensin II (AngII) into angiotensin-(1-9) (Ang-(1-9)) and angiotensin-(1-7) (Ang-(1-7)), respectively, as well as participates in tryptophan uptake in the gut and in COVID-19 infection. Our aim was to investigate the metabolic effect of ACE2 deletion in young adults and elderly mice under conditions of high calorie intake. Male C57Bl/6 (WT) and ACE2-deficient (ACE2^-/y^) mice were analyzed at the age of 6 and 12 months under standard diet (StD) and high-fat diet (HFD). Under StD, ACE2^-/y^ showed lower body weight and fat depots, improved glucose tolerance, enhanced insulin sensitivity, higher adiponectin, and lower leptin levels compared to WT. This difference was even more pronounced after HFD in 6-month-old mice, but, interestingly, it was blunted at the age of 12 months. ACE2^-/y^ presented a decrease in adipocyte diameter and lipolysis, which reflected in the upregulation of lipid metabolism in white adipose tissue through the increased expression of genes involved in lipid regulation. Under HFD, both food intake and total energy expenditure were decreased in 6-month-old ACE2^-/y^ mice, accompanied by an increase in liquid intake, compared to WT mice, fed either StD or HFD. Thus, ACE2^-/y^ mice are less susceptible to HFD-induced obesity in an age-dependent manner, as well as represent an excellent animal model of human lipodystrophy and a tool to investigate new treatments.

## 1. Introduction

Angiotensin-converting enzyme type 2 (ACE2; EC 3.4.17.23) is a pivotal component of the renin-angiotensin system (RAS), which functions as a peptidase that hydrolyzes angiotensin I (AngI) and angiotensin II (AngII) into angiotensin-(1-9) (Ang-(1-9)) and angiotensin-(1-7) (Ang-(1-7)), respectively [1,2]. This enzyme is, therefore, considered as a negative regulator of the RAS for decreasing the levels of AngII through the angiotensin type 1 receptor and increasing Ang-(1-7) [3]. These actions confer ACE2 a protective role in cardiovascular diseases [3,4,5]. Furthermore, ACE2 was recently demonstrated to be the receptor for the entry of the SARS-CoV-2 virus into cells, leading to the emergence of COVID-19 [6]. The mechanism of entry of SARS-CoV-2 into cells is the binding of the viral spike protein to ACE2 in several cell types, including the epithelium of the respiratory system [6,7]. Increased susceptibility to SARS-CoV-2 infection in patients with chronic diseases, such as obesity [8] and diabetes [8,9], is related to increased ACE2 expression in these diseases [10,11].

The pathological accumulation of adipose tissue is one of the major risk factors for cardiovascular disease [12,13]. In this context, studies also involving the RAS have largely been conducted to better understand the metabolic and cardiovascular pathophysiology, both in animals [14,15] and humans [12,16]. Thus, considering the severity of metabolic diseases, the ACE2 pathway of the RAS is an important pathway to be mechanistically investigated. The role of the ACE2/Ang-(1-7)/Mas axis in homeostasis is widely demonstrated in several metabolic studies [17,18,19]. ACE2 inhibits liver fibrosis [19], protecting against the development of diabetic retinopathy [20] and acute lung failure [21], whereas the genetic deletion of this enzyme leads to endothelial dysfunction and redox imbalance in mice [22], as well as hepatic steatosis [18]. In fact, ACE2 has been shown to exert pleiotropic actions [3,4,5,22,23] and the impairment in its expression or activity leads to, among others, cardiac [3], vascular [24,25], and renal disorders [26].

ACE2 has also been described as a regulator of dietary amino acid transport in the gut [23,27]. In mice, the deficiency results in high susceptibility to intestinal inflammation induced by epithelial damage, since the uptake of the essential amino acid tryptophan via the B^0^AT1/ACE2 transport pathway on the small intestinal epithelial cells is impaired in these animals, resulting in dysregulation of the gut microbiota and propensity to inflammation [23]. Moreover, our group has shown that this decreased uptake of tryptophan, the amino acid precursor of serotonin (5-HT), causes reduction in 5-HT levels in the blood and brain in ACE2-deficient mice [28]. Moreover, tryptophan hydroxylase (TPH) 1-deficient mice, animals with low peripheral 5-HT, are protected from high-fat diet-induced obesity [29]. Under HFD conditions, the inhibition of 5-HT synthesis in mice reduced body weight gain, improved glucose tolerance, and decreased lipogenesis in white adipose tissue [30].

Despite their relevance, the precise roles of ACE2 in lipid metabolism and adipose tissue remain unclear. Considering the emersion of metabolic diseases and the multifunctionality of ACE2, as well as its emerging role in the onset and severity of COVID-19, our aim was to investigate the effect of HFD during the lifespan in ACE2-deficient mice, especially regarding systemic metabolism of lipids and glucose. Therefore, the hypothesis was that ACE2 interferes with glycemic and lipid pathways due to its key role in homeostasis, and ACE2^-/y^ mice may present a different profile from high-fat diet-induced obesity. In this study, we showed that the deletion of ACE2 decreases the white fat depots and the susceptibility to high-fat diet-induced obesity in young adults, but not in elderly mice. In addition, ACE2 deficiency improved glycemic metabolism, a central mechanism controlling metabolic homeostasis.

## 2. Results

### 2.1. ACE2 Deficiency Decreases White Fat Depots and the Susceptibility to High-Fat Diet-Induced Obesity

Under StD, ACE2^-/y^ showed a lower BW, total fat, and WAT index at 6 and 12 months of age compared to WT (Figure 1A–C, respectively). These differences were even more pronounced after HFD in 6-month-old mice; however, surprisingly, they disappeared at the age of 12 months. In addition, ACE2^-/y^ displayed a decrease in white adipocyte diameter in 6-month-old mice under both HFD and StD (Figure 1D).

The significant decrease in fat depots led us to evaluate whether lipolysis was modified. In vivo, ACE2^-/y^ mice presented lower NEFA release after selective β3-adrenoceptor agonist stimulation under both StD and HFD when compared to WT mice (Figure 2A). Nonetheless, in vitro, these differences were no longer observed (Figure 2B). To understand the mechanism of altered lipolysis in ACE2^-/y^, we evaluated the expression of genes involved in this pathway in WAT. Under StD, the mRNA levels for *lipoprotein lipase* (*LPL*) and *adrenergic β3-receptor* (*Adrβ3*) (Figure 2C,D) did not differ between the groups in 6-month-old mice but were significantly decreased in 12-month-old ACE2^-/y^ mice. In HFD condition, *LPL* mRNA showed no difference between the groups. However, interestingly, the *Adrβ3* mRNA levels were significantly increased in ACE2^-/y^, both in 6 and 12-month-old mice. Consistent with this, in WAT, the expression of *hormone-sensitive lipase* (*HSL*), a lipase that hydrolyzes stored triglycerides to free fatty acids, was significantly increased in 6-month-old ACE2^-/y^ mice in both diet conditions, but not in 12-month-old animals (Figure 2E).

As expected, according to the amount of WAT, ACE2^-/y^ mice presented higher adiponectin and lower leptin levels in plasma, with the same holding true for the gene expression of *leptin* and *adiponectin* in WAT compared to WT. These differences were again blunted at the age of 12 months under HFD (Figure 2F–I).

The percentage of muscle and water was evaluated to assess whether changes in these parameters contributed to the decrease in BW presented by ACE2^-/y^ mice. Under StD, the results showed no difference in the percentage of muscle between 6-month-old ACE2^-/y^ mice and WT, but a significant increase was observed in 12-month-old ACE2^-/y^ mice compared to WT (Table 1). Nevertheless, under HFD, the percentage of muscle in 6-month-old ACE2^-/y^ mice was greater in comparison to WT, a fact not observed in 12-month-old ACE2^-/y^ mice under HFD. The data further demonstrate that at 6 months, ACE2^-/y^ under HFD compared to ACE2^-/y^ under StD showed less muscle mass reduction than at 12 months. In terms of the percentage of water, no differences were observed between ACE2^-/y^ mice compared to WT groups (Table 1).

Under StD, the plasma lipid profile showed no difference in TG and TCOL levels in 6- and 12-month-old ACE2^-/y^ mice compared to WT (Table 1). Notwithstanding, ACE2 deletion induced a significant decrease in NEFA levels in 6-month-old mice under isocaloric conditions, but not under HFD (Table 1). No differences were observed in NEFA levels between 12-month-old groups. As previously published by our group [18], under StD, ACE2^-/y^ mice presented hepatic steatosis with high deposition of NEFA in the liver, which explains its lower systemic concentration.

### 2.2. ACE2 Deficiency Improved Glucose Control and Insulin Signaling

Glycemic profile was evaluated by fasting glucose and insulin measurements, as well as by assessing glucose tolerance and insulin sensitivity [31]. Fasting glucose levels showed no difference between ACE2^-/y^ and WT groups at both ages under StD and HFD. However, compared with StD, the HFD increased fasting glucose levels in WT and ACE2^-/y^ (Table 1). The levels of fasting insulin in plasma showed no significant difference between the groups under StD and HFD (Table 1).

ACE2 deletion increased glucose tolerance at the ages of 6 and 12 months (Figure 3A,B, respectively) and enhanced insulin sensitivity under StD (Figure 3C,D, respectively). Even after HFD, 6-month-old ACE2^-/y^ mice continued to present better glucose tolerance and insulin sensitivity compared to WT mice. Nevertheless, in 12-month-old ACE2^-/y^ mice, these improvements were blunted after HFD.

The expression of genes involved in glucose and insulin regulation was measured in WAT and muscle. Under StD, 6-month-old ACE2^-/y^ mice showed augmented levels of *insulin receptor* (*IR*) mRNA in both organs, but with statistical significance only in the muscle (Figure 3E,F). However, after HFD, *IR* expression significantly increased in WAT, but not in the muscle. Meanwhile, 12-month-old ACE2^-/y^ mice showed no difference in *IR* mRNA in WAT and muscle under StD and HFD (Figure 3E,F, respectively). The levels of *glucose transporter type 4* (*GLUT4*) mRNA in WAT were not different between groups at both ages under StD (Figure 3G). However, such levels were significantly higher in WAT of 6-month-old ACE2^-/y^ after HFD but not in 12-month-old ACE2^-/y^ (Figure 3G). In muscle, no difference in *GLUT4* expression was observed between groups (Figure 3H). These results partly explain the better glucose tolerance and insulin sensitivity in 6-month-old ACE2^-/y^ animals, even after HFD ingestion, which was lost in the 12-month-old animals.

### 2.3. ACE2 Deficiency Leads to Impairment in Energy Balance

The balance between energy intake and expenditure reflects the control of BW and fat depots. Both parameters were decreased in 6-month-old ACE2^-/y^ compared to WT mice after HFD but not StD. On the other hand, liquid intake was increased in 6-month-old ACE2^-/y^ animals compared to WT under StD and HFD (Table 1).

### 2.4. ACE2 Deficiency Did Not Alter Antioxidant Enzyme Activity in WAT

To evaluate the involvement of antioxidant enzymes in the mechanism of resistance to high-fat diet-induced obesity, the activity of SOD and CAT was measured and normalized by total protein concentration in WAT [32]. As observed in Figure 4, in 6-month-old animals under StD and HFD, no difference in SOD or CAT activity is found in ACE2^-/y^ compared to WT mice (Figure 4A,B, respectively).

## 3. Discussion

The present study demonstrates that the deletion of ACE2 leads to diminished BW and WAT depot, regardless of age, in addition to the lower susceptibility to HFD-induced obesity in young adult (but not elderly) mice, confirming our hypothesis that ACE2^-/y^ mice are protected from high-fat diet-induced obesity. The impact of elevated ACE2 expression in adipose tissue on health is still controversially discussed. This discussion has become even more evident with the emergence of COVID-19, since the novel coronavirus SARS-CoV-2 causes ACE/ACE2 balance disruption by using ACE2 as a receptor to enter host cells [6]. In COVID-19 patients, pre-existing comorbidities, such as obesity, contribute to severe forms of the disease and, consequently, higher mortality rates [8,33]. The present study demonstrated that ACE2 deletion in mice decreased the susceptibility to HFD-induced obesity and improved glucose homeostasis, suggesting a prominent role of ACE2 in the emergence of metabolic disorders. In consistency with our findings, Gupte and coauthors [34] showed that ACE2 is upregulated in adipose tissue of rodents with HFD-induced obesity.

The decrease in fat deposition and adipocyte diameter presented by ACE2^-/y^, even after HFD, was reflected in the downregulated lipolysis in vivo. Nevertheless, under StD, there was an increase in *HSL* at 6 months and a reduction in *LPL* at 12 months. Meanwhile, under HFD, there was an augmentation *in adrenergic β3 receptors* and in *HSL*, which are lipolysis stimulators. In that regard, Cai and colleagues [35] demonstrated that the infusion of AngII for 2 weeks in C57BL/6 mice induced body and white adipose tissue weight loss by promoting lipolysis by activating the AMPK signaling pathway, which partially explains the results presented by ACE2^-/y^ mice as a function of possible increased AngII levels. Furthermore, it was postulated that catalase activity could be involved in the mechanism, since it has been shown that 3-Amino-1,2,4-Triazole, a catalase inhibitor, induces quick fat loss in mice with high fat-induced metabolic syndrome [36]. However, the catalase activity in WAT was not altered in ACE2^-/y^ mice in either StD or HFD conditions. Therefore, it is postulated that the marked decrease in adipose tissue deposition in ACE2^-/y^ mice could reflect lipodystrophy, considering that there is a greater uptake of NEFA by the liver [18] and, consequently, their lower systemic levels. In this context, in the long term, it was reflected in the decrease in lipolysis due to the decrease in WAT deposition and less size expansion of adipocytes under HFD. WAT stores energy and is considered a complex and dynamic endocrine organ with key metabolic, cardiac, and vascular regulatory functions [37,38]. A disorder in this tissue triggers reduced production of adipocytokines, such as adiponectin, and upregulation of the expression of leptin, tumor necrosis factor-α (TNF-α), and interleukin-6 (IL-6), promoting insulin resistance in association with increased visceral adiposity [14,39]. Systemic adiponectin levels are inversely correlated with the amount of WAT, and the opposite is observed for leptin levels [40]. This is consistent with the observation that the lower WAT index presented by ACE2^-/y^ resulted in an increase in adiponectin and decrease in leptin levels in plasma, even after HFD consumption at 6 months of age. Among its several biological functions, adiponectin plays an anti-inflammatory and insulin sensitizing role [41]. Meanwhile, in *knockout* mice, adenoviral adiponectin overproduction ameliorates diet-induced insulin resistance [42]. Additionally, Combs et al. [43] identified a dominant mutation in the collagenous domain of this adipocytokine that elevated its circulating levels in mice and also improves insulin sensitivity [43]. In the present work, the increase in systemic levels of this hormone in ACE2^-/y^ mice contributed to a better insulin sensitivity and glucose tolerance profile, even concomitantly with HFD consumption. Contributing to this favorable glycemic profile, ACE2^-/y^ mice presented higher expression of *IR* in the muscle under StD, and of *IR* and *GLUT4* in WAT under HFD. It is important to highlight that after 60 min of the intraperitoneal application of insulin in 6-month-old ACE2^-/y^ mice during the test, insulin sensitivity was so intense that glucose reached extremely low levels and, to avoid hypoglycemic shock, glucose (5% in saline, 10 µL·g^−1^·BW^−1^) was applied subcutaneously at the end of the experiment. Taken together, these data suggest a greater uptake of glucose in these important metabolic organs.

Leptin, in turn, plays a key role in regulating energy intake and expenditure, and the levels of this protein in the blood positively correlate with adipose mass [44]. Thus, one could expect increased food intake of ACE2^-/y^ mice, since the arcuate nucleus of the hypothalamus recognizes the lower circulating levels of leptin, ultimately stimulating food intake and decreasing energy expenditure [45]. As expected, under HFD, 6-month-old ACE2^-/y^ mice showed lower energy expenditure, but, surprisingly, they presented decreased food intake, which, in part, is in accordance with the lower fat depots and the lower susceptibility to HFD-induced obesity in these mice. However, since ACE2 deficiency in mice results in high susceptibility to the dysregulation of the gut microbiota and intestinal inflammation, induced by epithelial damage [23], this system may contribute to lower food intake, despite the decreased levels of leptin. Furthermore, ACE2^-/y^ mice showed an increase in water intake independent of the diet, probably induced by the high levels of AngII, since it stimulates thirst [46].

The phenotype of ACE-deficient mice, exhibiting lower BW, body fat, and plasma leptin, is similar to that of ACE2^-/y^ mice found in the present work. Jayasooriya and colleagues, however, did not associate these findings with decreased food intake, since no differences were detected in daily food intake in ACE^-/-^ mice, but they associated their observations to a high energy expenditure related to an increased metabolism of fatty acids in the liver, instead [47].

Despite the association between advanced age and consumption of HFD, conditions that are already known to favor the onset of metabolic diseases [48], the deletion of ACE2 at the age of 12 months did not change BW, fat deposition, or plasma lipid and glucose homeostasis, demonstrating a delayed effect of HFD in ACE2^-/y^ mice, suggesting that ACE2 in an old age no longer interferes with these parameters as observed at younger ages, and that the mechanism of protection against HFD-induced obesity could be age-dependent. In addition to this, since both ACE2 deficiency and obesity have been characterized as an accelerated aging phenotype [49], both conditions together could explain the reason why some of the beneficial effects observed in ACE2^-/y^ disappear in the 12-month age.

In addition to the RAS, ACE2 deficiency also interferes with the 5-HT system [28]. Considering that 5-HT has localized effects on adipose tissues, specifically on the lipogenesis of WAT [30], as well as the fact that the decrease in peripheral 5-HT confers protection from HFD-induced obesity [29,30], and that ACE2^-/y^ mice have less 5-HT [28] due to the defective intestinal absorption of the 5-HT precursor amino acid, tryptophan [23], it is plausible to consider that their protection against HFD-induced obesity is mediated by decreased levels of 5-HT. Singer and colleagues [27] also demonstrated that the intestinal absorption of amino acids was defective in ACE2-deficient mice. The authors confirmed the absorptive defect by the intensified increase in L-tryptophan and other neutral amino acids in the lumen of the ileum, associated with decreased levels of these amino acids in the plasma and muscle of ACE2-deficient mice.

In contrast to our findings, other authors [50,51] previously reported that ACE2^-/y^ mice, under StD, had normal insulin sensitivity and glucose tolerance, and after HFD these parameters were even impaired. Furthermore, unlike Takeda et al. [50] and Niu et al. (2008), our ACE2^-/y^ mice showed lower BW, which, together with the lower fat depots and the insulin sensitizing action exerted by the elevated adiponectin levels, resulted in the improvement of glucose and insulin homeostasis. These results were the same in three different experiments performed by our group. We can only speculate about the reasons for the discrepant results between our study and the previous ones. The two other groups used different ACE2-deficient animals than ours. Takeda et al. deleted exon 3 of the *ACE2* gene, which leaves the collectrin-like domain intact. There may be still residual expression of a shortened ACE2 in the small intestine, which lacks the enzymatic function but retains the transporter role. This would allow for tryptophan transport and, thereby, the mice may have normal 5-HT levels, while our strain has clearly lost this function [28]. However, in the study by Niu and colleagues, the exact targeting strategy for the *ACE2* gene is not described, and tryptophan or serotonin levels are also not given. Recently, a short isoform of ACE2 was described in humans transcribed from a promoter in intron 9 and also coding for a protein which retains the collectrin-like domain [52]. However, this short ACE2 isoform has not yet been reported to exist in mice, and thus, it remains unclear whether it may be expressed in some of the different ACE2-deficient mouse models. Nevertheless, we propose that in our mice, the protective effects mediated by the lack of 5-HT overweigh the deleterious effects of the lack Ang-1-7, and this may explain the difference between our results and other studies. Despite their relevance, the precise role of ACE2 in lipid metabolism and adipose tissue remains unclear.

Hence, despite the beneficial effect shown in the present work (summarized in Figure 5), previous studies by our group demonstrated that ACE2 deletion leads to endothelial dysfunction and vascular oxidative stress [22], as well as steatosis and impaired insulin signaling in the liver [18]. Taken together, given its pleiotropic action, our data suggest that the level and location of ACE2 expression differentially control important hormone systems, such as the RAS and the 5-HT system, to ensure cardiovascular and metabolic homeostasis.

## 4. Materials and Methods

### 4.1. Animals and Experimental Procedures

C57Bl/6 (WT) and ACE2-deficient (ACE2^-/y^) mice [3] were used in this study in accordance with the “Guide for the Care and Use of Laboratory Animals” (NIH publication 86–23, 1996). To reduce the hormonal influence in the analyses, male mice were used. To ensure the deletion of ACE2 in mice, genotyping was carried out by Polymerase Chain Reaction (PCR) using an ear clip. All ACE2^-/y^ mice used in this study had gene deletion confirmed. Throughout the study, the mice were kept on a 12 h light/dark cycle and fed ad libitum, with controlled humidity and environmental temperature (21–22 °C). In the experimental design, WT and ACE2^-/y^ mice were placed on a high-fat diet (HFD). For analysis at an advanced age, and another group of WT and ACE2^-/y^ mice were placed on HFD for 20–22 weeks before 12 months of age. The standard diet (StD, V1536-kJ: 10% fat, 23% protein, and 67% carbohydrates) and HFD (E15744-347-kJ: 45% fat, 20% protein, and 35% carbohydrates), both diets from SSNIFF^®^ (Arnsberg, Germany), were used.

### 4.2. Glucose Tolerance and Insulin Sensitivity Tests

The glucose tolerance was evaluated after an overnight fast (12 h). The animals were weighed, and the blood was collected from the tail vein for measuring glycaemia (Merck^®^, Hesse, Germany) before intra-peritoneal (i.p.) injection of 2 g glucose per kg of body weight. Subsequently, the glucose levels were measured at 15, 30, 60, 90, and 120 min. An insulin sensitivity test was assessed with the animal in the fed state. Similar to the aforementioned test, the animals were weighed, and the blood was collected for measuring glucose before the injection of 0.75 units insulin.kg^−1^, i.p. (Huminsulin, Lilly^®^, Indianapolis, IN, USA) and after injection at 15, 30, 45, and 60 min [31]. For all blood glucose measurements, a handheld glucometer was used (Accu-Chek AVIVA*,* Roche^®^, Mannheim, Germany).

### 4.3. Metabolic Cage System

The mice were adapted in metabolic cages for 2 days. After this, the measurements of 24 h activity, energy expenditure, food, and water consumption were performed using a combined indirect calorimetry system (TSE Systems GmbH, Bad Homburg, Germany) for three days. The results were expressed as the average of three days.

### 4.4. Body Composition Analysis

The body composition was performed in unanesthetized mice using the non-invasive and non-destructive Bruker^®^ minispec LF90II nuclear magnetic resonance (NMR) technology analyzer mq10 (Bruker Optics, Billerica, TX, USA). In the test, mice were individually placed in small tubes and then inserted into an NMR analyzer to provide quantitative analysis of tissue and body fluids. The body weight (BW), total fat, muscle, and water were recorded.

### 4.5. Lipolysis In Vivo and In Vitro

For the assessment of lipolysis in vivo, mice were i.p. administered with 1 mg per kg body weight of the selective β3-adrenoreceptor agonist CL-316,243 hydrate (C5976; Sigma-Aldrich^®^, Hanover, Germany). Non-esterified fatty acids (NEFAs) were measured in blood collected from the tail vein before (basal) and 15 and 30 min after the injection.

Lipolysis was assessed in vitro in white adipose tissue collected after euthanasia. The tissue was incubated in a culture medium (DMEM, Gibco^®^ 11880; Hesse, Germany) in a bath (37 °C; 95% O_2_; 5% CO_2_) for 30 min, and then the β3-adrenoceptor agonist CL-316,243, 0.1 mM, was added to the medium. The free fatty acids were measured in the medium at 0 and 180 min of incubation and normalized by the amount of fat used.

### 4.6. Euthanasia and Ex Vivo Experiments

At the end of the experimental design, in a fasted state, all mice were anesthetized (100 mg·kg^−1^ ketamine, 10 mg·kg^−1^ xylazine, i.p.) and euthanized by exsanguination through cardiac puncture of the right ventricle. The whole blood was collected and centrifuged (4000 rpm for 10 min), and the plasma was separated. The animals were perfused with heparinized saline, and, in sequence, white adipose tissue (WAT) and gastrocnemius muscle were carefully removed and weighed. All tissues were immediately frozen in dry ice and stored at −80 °C until the analyses were performed. WAT index was calculated using the following formula: WAT index (%) = Epididymal fat + Perirenal fat)/(Body Weight × 100.

### 4.7. Biochemical Analyses

For measurement of insulin, leptin, and adiponectin, ELISA assays were used (Millipore^®^, Hesse, Germany) according to the manufacturers’ instructions. A Wako NEFA kit was used to measure plasma and liver NEFA concentrations (Wako Chemicals GmbH^®^, Neuss, Germany). Triglycerides (TG) and total cholesterol (TCOL) levels were assayed using commercial kits (Labtest^®^, Belo Horizonte, Brazil), following the manufacturers’ instructions with adaptations for microplates. All measurements were performed using a TECAN^®^ Infinite 200 PRO plate reader (Zurique, Switzerland).

### 4.8. Superoxide Dismutase (SOD) and Catalase (CAT) Activities

Total SOD activity was assessed in WAT by commercial colorimetric kit (Sigma Aldrich, Hanover, Germany) and read in a microplate reader (Thermo Scientific, Software 2.4 Multiskan Spectrum, Vantaa, Finland) at 450 nm. The obtained values were normalized by total protein concentration [32] in the WAT and expressed as IU⋅protein^−1^. Catalase activity was measured according to Xu and coauthors [15], being expressed as μmol/min/mL⋅protein^−1^, mg [32].

### 4.9. Adipose Tissue Histological Analysis

Small fragments of WAT were fixed in 4% buffered formaldehyde, embedded in paraffin, and sectioned at 10 µm. Adipose tissue was stained with hematoxylin and eosin (H&E) staining in order to determine the adipocyte diameter [18]. The sections from each animal were histologically examined in light microscope (Keyence^®^ microscope, BZ 9000, Osaka, Japan), photographed, and analyzed using the “BZ-9000 Generation II Analyzer” image processing software (Keyence^®^ BZ 9000 Software, Ver.2.1 (BZ-H2A/H2AE), Osaka, Japan).

### 4.10. Quantitative Real-Time PCR

Total RNA was isolated from WAT and muscle tissues by using trizol (TRizol^®^ Reagent, Darmstadt, Germany) and cleaned using the RNeasy Mini kit (Qiagen, Hilden, Germany) to ensure high-quality total RNA extraction. RNA was quantified by using spectrophotometry (NanoDrop^®^, München, Germany), and 1 μg was used for the synthesis of cDNA (Reverse Transcriptase–Invitrogen^®^) performed using M-MLV (S1000™ Thermal Cycler, Bio-Rad^®^ München, Germany). The reaction product was amplified using the GoTaq qPCR Master Mix (Promega^®^; Mannheim, Germany) by real-time quantitative PCR (ABI 7900HT Real-Time PCR System-Applied Biosystems, Darmstadt, Germany). mRNA was quantified as a relative value compared with an internal reference, *Glyceraldehyde 3-phosphate dehydrogenase* (*GAPDH*). The expression levels were obtained from the cycle threshold (Ct) associated with the exponential growth of the PCR products. Quantitative values for mRNA expression were obtained by the parameter 2^−ΔΔCt^, in which ΔCt represents the subtraction of the *GAPDH* Ct values from the others. The sequence and length of all primers used for real-time quantitative PCR (amplicons between 100–150 bp) are listed in Table 2.

### 4.11. Statistical Analysis

Data are expressed as mean ± standard error of the mean (S.E.M), and *p* ˂ 0.05 was considered statistically significant. Analyses to compare multiple groups were performed by using one-way ANOVA, to evaluate an independent variable, or two-way ANOVA followed by Bonferroni post-test, to analyze two independent variables (GraphPad Prism^®^ 5.0, San Diego, CA, USA).

## 5. Conclusions

In summary, ACE2^-/y^ mice are less susceptible to HFD-induced obesity in an age-dependent manner. Moreover, our data suggest that to ensure cardiovascular and metabolic homeostasis, ACE2 controls important hormone systems, such as the RAS and the 5-HT system, as well as represents an excellent animal model of human lipodystrophy and a research tool to investigate new treatments, as well as to better comprehend the signaling pathways involved in metabolic diseases.

## Figures and Tables

**Figure 1 ijms-25-09515-f001:**
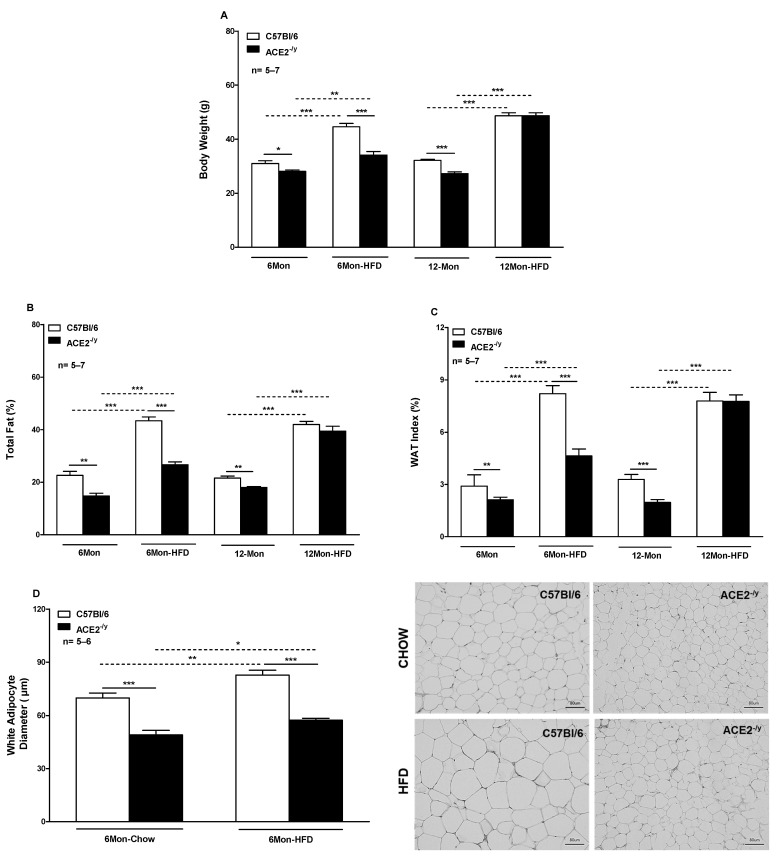
(**A**) Body weight; (**B**) Total fat; (**C**) WAT index of C57Bl/6 and ACE2^-/y^ mice at the age of 6 and 12 months under standard diet (StD, chow) and high-fat diet (HFD); (**D**) White adipocyte diameter of C57Bl/6 and ACE2^-/y^ mice at the age of 6 months (Mon) under StD and HFD. Each bar graph represents the mean ± SEM. ANOVA (one way): * *p* < 0.05; ** *p* < 0.01; *** *p* < 0.001 KO vs. WT.

**Figure 2 ijms-25-09515-f002:**
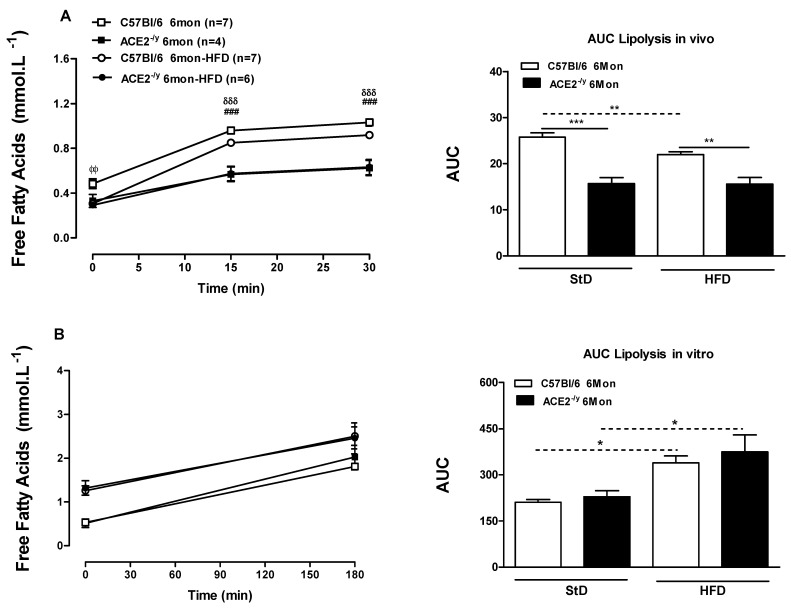
(**A**) Lipolysis in vivo; (**B**) Lipolysis in vitro; (**C**) Relative expression of *lipoprotein lipase* (*LPL*) in white adipose tissue (WAT); (**D**) Relative expression of *Adrβ3 receptor* in WAT; (**E**) Relative expression of *hormone-sensitive lipase* (*HSL*) in WAT; (**F**) Relative expression of *Adiponectin* in WAT. (**G**) Adiponectin levels in plasma; (**H**) Relative expression of *Leptin* in WAT; (**I**) Leptin levels in plasma of C57Bl/6 and ACE2^-/y^ mice at the age of 6 and 12 months (Mon) under standard diet (StD) and high-fat diet (HFD). Each point on the graph represents the mean ± SEM. ANOVA (two ways): * *p* < 0.05; ** *p* < 0.01 KO vs. WT. Each bar graph represents the mean ± SEM. ANOVA (one way): * *p* < 0.05; ** *p* < 0.01; *** *p* < 0.001 KO vs. WT. ^###^ ACE2^-/y^ StD vs. WT StD; ^ΦΦ^ WT HFD vs. WT StD; ^δδδ^ ACE2^-/y^ HFD vs. WT HFD.

**Figure 3 ijms-25-09515-f003:**
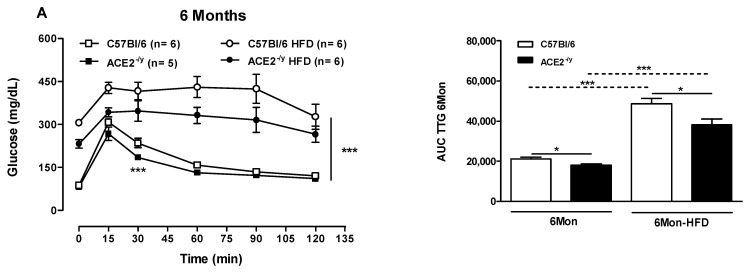
Glycemic curve and area under curve (AUC) of the glucose tolerance (**A**,**B**) and insulin sensitivity (**C,D**) of C57Bl/6 and ACE2^-/y^ mice at the age of 6 (**A**,**C**) and 12 (**B**,**D**) months (Mon) under standard (StD) and high-fat diet (HFD). (**E**,**F**) *Insulin receptor* (*IR*) mRNA expression in white adipose tissue (WAT) and muscle, respectively, and (**G**,**H**) *GLUT4* mRNA expression in WAT and muscle, respectively, in 6-month-old animals under StD and HFD. Each point on the graph represents the mean ± SEM. ANOVA (two ways): * *p* < 0.05, ** *p* < 0.01, *** *p* < 0.001 KO vs. WT. Each bar graph represents the mean ± SEM. ANOVA (one way): * *p* < 0.05; ** *p* < 0.01; KO vs. WT.

**Figure 4 ijms-25-09515-f004:**
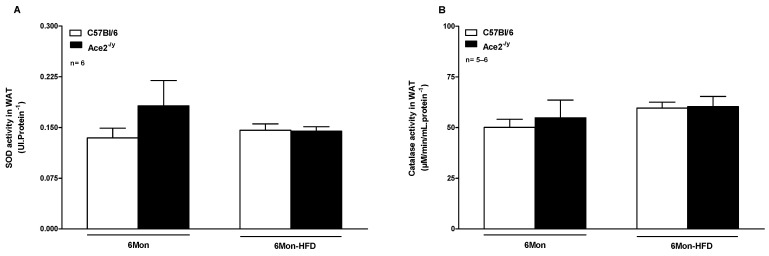
(**A**) Superoxide Dismutase (SOD) and (**B**) Catalase (CAT) activities in white adipose tissue (WAT) of C57Bl/6 and ACE2^-/y^ mice at the age of 6 months (Mon) under standard diet (StD) and high-fat diet (HFD). Each bar graph represents the mean ± SEM. ANOVA (one way).

**Figure 5 ijms-25-09515-f005:**
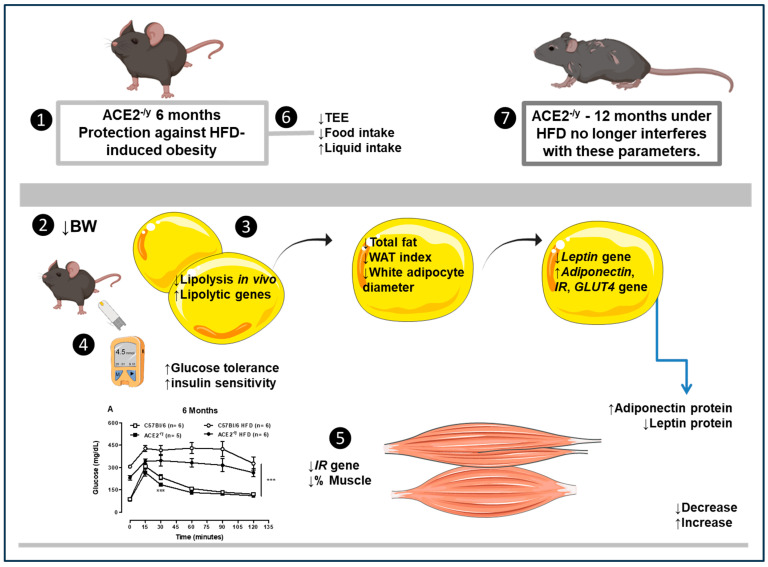
❶ The 6-month-old ACE2^-/y^ mice are protect against HFD-induced obesity. ❷ The animals presented a decrease in body weight (BW), ❸ total fat, white adipose tissue (WAT) index, white adipocyte diameter, and *leptin* gene in WAT; decrease in lipolysis in vivo, increase in lipolytic (*HSL, Adrβ3*) and *adiponectin* genes in WAT. ❹ Systemically, these effects reflected in the increase in glucose tolerance, insulin sensitivity, and adiponectin, as well as a decrease in leptin protein. ❺ In the gastrocnemius muscle, it is observed that there is a decrease in *insulin receptor* (*IR)* gene and % muscle, which possibly reflects in ❻ the decrease in total energy expenditure (TEE) and food intake detected in ACE2^-/y^ mice. Regarding the liquid intake, ACE2^-/y^ mice present an increase, probably induced by the high levels of AngII, since it stimulates thirst. ❼ The deletion of ACE2 at the age of 12 months did not change BW, fat deposition, or plasma glucose homeostasis under HFD, demonstrating a delayed effect of HFD in ACE2^-/y^ mice, suggesting that ACE2 in an old age no longer interferes with these parameters as observed in younger ages, and that the mechanism of protection against HFD-induced obesity could be age-dependent. *** *p* < 0.001.

**Table 1 ijms-25-09515-t001:** Metabolic parameters, food intake, and total energy expenditure of C57Bl/6 and ACE2^-/y^ mice at the age of 6 and 12 months under standard (StD) and high-fat diet (HFD).

Parameters	6 Months				12 Months			
	Standard Diet		High-Fat Diet		Standard Diet		High-Fat Diet	
	C57Bl/6	ACE2^-/y^	C57Bl/6	ACE2^-/y^	C57Bl/6	ACE2^-/y^	C57Bl/6	ACE2^-/y^
Muscle %	67.77 ± 1.54	72.85 ± 1.78	48.11 ± 1.40 ^c^	64.85 ± 1.12 ^bd^	69.50 ± 0.74	72.84 ± 0.27 ^a^	49.92 ± 1.17 ^c^	51.85 ± 1.88 ^b^
Water %	7.15 ± 0.11	7.34 ± 0.4	6.47 ± 0.13 ^c^	6.43 ± 0.18 ^b^	6.54 ± 0.19	6.71 ± 0.12	6.16 ± 0.11	6.54 ± 0.14
Triglyceride (mg·dL^−1^)	22.02 ± 1.9	16.57 ± 2.3	26.9 ± 2.2	29.0 ± 1.7 ^b^	33.9 ± 3.3	32.9 ± 7.4	45.8 ± 6.0	44.9 ± 3.9
Cholesterol (mg·dL^−1^)	36.97 ± 2.1	45.23 ± 4.4	126.3 ± 4.7 ^c^	140.7 ± 10.2 ^b^	68.1 ± 5.7	73.3 ± 12.6	173.6 ± 11.1 ^c^	210.3 ± 27.6 ^b^
NEFA (mmol·L^−1^)	0.29 ± 0.03	0.14 ± 0.01 ^a^	0.49 ± 0.05 ^c^	0.49 ± 0.05 ^b^	0.48 ± 0.1	0.27 ± 0.07	0.42 ± 0.04	0.42 ± 0.09
Glucose (mg·dL^−1^)	121.6 ± 8.8	103.5 ± 7.0	153.0 ± 9.3 ^c^	165.2 ± 7.8 ^b^	138.8 ± 9.0	130.6 ± 6.6	187.8 ± 6.7 ^c^	176.8 ± 6.9 ^b^
Insulin (pg·dL^−1^)	179.2 ± 12.2	175.7 ± 4.9	238.7 ± 25.8	184.1 ± 4.4	303.5 ± 6.8	301.2 ± 9.2	333.0 ± 17.5	331.1 ± 17.3
Food intake (g·day^−1^)	5.12 ± 0.44	4.64 ± 0.15	4.57 ± 0.37	3.34 ± 0.18 ^bd^	ND	ND	ND	ND
Liquid intake (g·day^−1^)	3.00 ± 0.3	4.27 ± 0.1 ^a^	2.61 ± 0.2	4.68 ± 0.4 ^d^	ND	ND	ND	ND
Total Energy Expenditure (kcal·h^−1^)	0.55 ± 0.03	0.49 ± 0.01	0.58 ± 0.01	0.48 ± 0.01 ^d^	ND	ND	ND	ND

Each value represents the mean ± SEM. ANOVA (one way). ^a^ *p* < 0.05 ACE2^-/y^ StD vs. C57Bl/6 StD. ^b^ *p* < 0.05 ACE2^-/y^ HFD vs. ACE2^-/y^ StD. ^c^ *p* < 0.05 C57Bl/6 HFD vs. C57Bl/6 StD. ^d^ *p* < 0.05 ACE2^-/y^ HFD vs. C57Bl/6 HFD. ND: not defined.

**Table 2 ijms-25-09515-t002:** Sequences of primer used for real-time quantitative PCR (amplicons between 100–150 bp).

Primers	Sequence Forward and Reverse (5′–3′) and Length (bp)
*Leptin*	F: CGTGTGTGAAATGTCATTGATCCT (24) R: GACACCAAAACCCTCATCAAGAC (23)
*Adiponectin*	F: GGAACTTGTGCAGGTTGGAT (20) R: CCTTCAGCTCCTGTCATTCC (20)
*Lipoprotein lipase*	F: AGTGGCCGAGAGCGAGAAC (19) R: CCACCTCCGTGTAAATCAAGAAG (23)
*Hormone-sensitive lipase*	F: ACGGATACCGTAGTTTGGTGC (21) R: TCCAGAAGTGCACATCCAGGT (21)
*Adrenergic β3 receptor*	F: GCTGACTTGGTAGTGGGACTC (21) R: TAGAAGGAGACGGAGGAGGAG (21)
*Glucose transporter type 4*	F: TGATTCTGCTGCCCTTCTGT (20) R: GGACATTGGACGCTCTCTCT (20)
*Insulin receptor*	F: CCACCAATACGTCATTCACAAC (22) R: GGGCAGATGTCACAGAATCAA (21)
*Glyceraldehyde 3-phosphate dehydrogenase*	F: CCATCACCATCTTCCAGGAG (20) R: GTGGTTCACACCCATCACAA (20)

## Data Availability

The data presented in this study are available on request from the corresponding authors.

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
