# Peer review of "ACE2 Knockout Mice Are Resistant to High-Fat Diet-Induced Obesity in an Age-Dependent Manner"

_ijms, 2024, doi:10.3390/ijms25179515_

Round 1

Reviewer 1 Report

Comments and Suggestions for Authors

Please add line number in the manuscript to convenient for reviewing.

In the Introduction, please avoid short paragraphs. The short paragraphs can be merged with corresponding paragraph.

In the Introduction, please add a hypothesis at the end of introduction. In addition, the aim of this study ...metabolic effect of high fat diet... should be specific, which metabolism.

In the Materials and Methods, I suggest that the expression of ACE2 in ACE2-deficient mice should measured in order to verify that the gene knockout was successful. Besides, the standard diet and high fat diet should be provided in a table.

The initial of each word in sub-title should be capitalized.

Table 1. Please provide Genebank ID and amplicon length of all genes.

Statistical analysis. Please specify which data using one and which data use two-way ANOVA. All the data are in accordance with normal distribution?

Results. In addition to mRNA expression, the protein expression of genes should be checked with western blotting analysis if the author still have samples.

Discussion. Please avoid short paragraphs. In addition, the author should cite more relevant research to elaborate the effects of ACE2 knockout on the glycometabolism and lipid metabolism in mice. The nutrients metabolism can be  separately elaborated.

Author Response

Comments 1: Please add line number in the manuscript to convenient for reviewing.

Response 1: The authors thank the detailed comments from this reviewer. We have added line numbers in the manuscript.

Comments 2: In the Introduction, please avoid short paragraphs. The short paragraphs can be merged with corresponding paragraph.

Response 2: Thank you for pointing this out. We agree with this comment and have merged paragraphs 2 and 3 in the introduction (lines 89-90).

Comments 3: In the Introduction, please add a hypothesis at the end of introduction. In addition, the aim of this study“...metabolic effect of high fat diet...”should be specific, which metabolism.

Response 3: We agree and thank you for your contribution. The hypothesis (lines 114-116) was added, as well as the aim was adjusted (line 112) to emphasize this point. By removing "metabolic" from the aim, we emphasize the effect of the high fat diet on the metabolism of lipids and glucose, mentioned at the end of the aim.

Comments 4: In the Materials and Methods, I suggest that the expression of ACE2 in ACE2-deficient mice should measured in order to verify that the gene knockout was successful. Besides, the standard diet and high-fat diet should be provided in a table.

Response 4: According to this suggestion, we added the information that all ACE2-/y mice used in this study had gene deletion confirmed by genotyping, which was carried out by Polymerase Chain Reaction (PCR) using an ear clip (lines 129-131). This experiment is routinely performed in our Laboratory to confirm genetic deletion in animals used in our studies. Regarding the standard and high-fat diets, both are from SSNIFF® (Soest, Germany). We emphasize in the text the kJ of fat, protein, and carbohydrate of each one, as well as the reference number of diets (lines 137-139).

Comments 5: The initial of each word in sub-title should be capitalized.

Response 5: The authors agree and thank your observation. We have corrected all subtitles.

Comments 6: Table 1. Please provide Genebank ID and amplicon length of all genes.

Response 6: We do not have Genebank ID, the primers used in the study were from a primer bank designed and tested with amplicon length between 100-150bp, as is routinely performed in our scientific group. This issue appears in the methods section (lines 232-235). In addition, we have added the length of the Forward and Reverse sequences in Table 1.

Comments 7: Statistical analysis. Please specify which data using one and which data use two-way ANOVA. All the data are in accordance with normal distribution?

Response 7: In Statistical analysis, we add that one-way ANOVA was used to evaluate an independent variable, and two-way ANOVA followed by Bonferroni post-test was used to evaluate two independent variables (lines 240-241). Furthermore, the captions of each figure also identify which tests were performed. According to the Kolmogorov-Smirnov test used to analyze the normal distribution in small sample sizes, all the data are in accordance with normal distribution, except the analysis of mRNA lipoprotein lipase expression.

Comments 8: Results. In addition to mRNA expression, the protein expression of genes should be checked with western blotting analysis if the author still have samples.

Response 8: Thank you for pointing this out. We agree with this comment. However, for logistical and financial reasons, it was not possible to evaluate western blotting analysis, unfortunately.

Comments 9: Discussion. Please avoid short paragraphs. In addition, the author should cite more relevant research to elaborate the effects of ACE2 knockout on the glycometabolism and lipidmetabolism in mice. The nutrients metabolism can be separately elaborated.

Response 9: Thank you for pointing this out. We agree with this comment and have merged paragraphs 1, 2, and 3 in the discussion (lines 381-382; 387-388). Furthermore, the authors added more information and another reference that helps to explain and justify our results (lines 456-462; 719-721).

Reviewer 2 Report

Comments and Suggestions for Authors

The authors observed delayed accumulation of adipose tissue in ACE2-/y mice on HFD rather than an 'age-dependent effect'

Is it possible that more water drinking interferes with eating in ACE2-/y mice?

Is there any translation of the findings to human obesity? 

The experiment is well described and presented. The discussion is fine but the conclusion (In summary, there is accumulating evidence indicating that ACE2-/y mice represent an excellent animal model of human lipodystrophy and a research tool to investigate new treatments, as well as to better comprehend the signaling pathways involved in metabolic diseases) is not truly coming from the study. 

Minore:

SD is the abbreviation used for standard deviation.  Please consider changing it.

Table 2 is too busy. Please consider the lower accuracy of the presented data.

Author Response

Comments 1: The authors observed delayed accumulation of adipose tissue in ACE2-/y mice on HFD rather than an 'age-dependent effect'

Response 1: The authors thank this reviewer's detailed comments and observation. To emphasize this point we added information on the delayed effect of HFD in ACE2-/y mice to complement the discussion (line 456).

Comments 2: Is it possible that more water drinking interferes with eating in ACE2-/y mice?

Response 2: The authors thank your question. About the question, possibly not. Regardless of the diet, the ACE2-/y mice showed increased liquid consumption. As observed, food intake was not different between WT and ACE2-/y mice in the standard diet condition, but it was lower in the ACE2-/y mice in the HFD condition, and in both situations, there was greater water consumption in the ACE2-/y mice.

Comments 3: Is there any translation of the findings to human obesity?

Response 3: The authors thank your question. Not yet. However, two years ago we started a case-control study to evaluate the role of ACE2 in human obesity.

Comments 4: The experiment is well described and presented. The discussion is fine but the conclusion (In summary, there is accumulating evidence indicating that ACE2-/y mice represent an excellent animal model of human lipodystrophy and a research tool to investigate new treatments, as well as to better comprehend the signaling pathways involved in metabolic diseases) is not truly coming from the study.

Response 4: Thank you for pointing this out. We agree with this comment. We have, accordingly, revised the conclusion to emphasize this point (lines 525-529) in line with the study results.

Comments 5: SD is the abbreviation used for standard deviation. Please consider changing it.

Response 5: Thank you for your observation. We changed the abbreviation for the standard diet to StD.

Comments 6: Table 2 is too busy. Please consider the lower accuracy of the presented data.

Response 6: Thank you for your observation. The page orientation of table 2 in the Word file is in "Landscape" format, due to the large amount of information. It is possible that the page has changed to "Portrait" format in the Journal edition and the table has been modified. Therefore, we will ask the Journal to keep the "Landscape" format so that the configuration of Table 2 does not change.

Reviewer 3 Report

Comments and Suggestions for Authors

Nunes-Souza et al. present a study demonstrating the effect of ACE2 KO in mice under high fat conditions, with the positive effect of ACE KO being diminished in an aging model. It’s certainly interesting work, highlighting an important mechanism that certainly has relevance in today’s society. Overall the enthusiasm is high for the work with minor thoughts included below.

Minor Thoughts:

1.       Was the White Adipocyte Diameter any different in the 12 month groups?

2.       Extra comma’s in first section of page 7. “stimulation under both SD and HFD when compared to WT mice.”

3.       When were the mice placed on the HFD and at what age? It’s unclear if the mice were placed on the HFD at 4ish months of age and then a subset were sacrificed at 6 months of age, with the rest remaining on the diet until they were 12 months. Or if mice were placed on the diet 20-22 weeks before 6 months and 12 months of age and then subsequently sacrificed at those ages. I would just clarify that point in the methods.

4.       Is the 90 minute data point for Figure 2B missing? It is mentioned in methods.

5.       Table 2 has the columns too close together and it’s quite challenging to read. Consider either highlighting different columns or provide greater separation.

6.       There is some data missing from the 12 months groups, namely Figure 4 and food intake/water and total energy expenditure in Table 2. Were these variables not measured or what was the rational in leaving these out?

7.       Consider the fact that both ACE2 deficiency and obesity have been characterized as an accelerated aging phenotype (PMID: 3547169, 23160880). When placed together, this could explain why some of the beneficial effects of ACE2 KO disappear in the 12 month age group. These concepts may warrant being included in the discussion and provide a rational for the differences in the age groups.

Author Response

Comments 1: Nunes-Souza et al. present a study demonstrating the effect ofACE2 KO in mice under high fat conditions, with the positive effect of ACE KO being diminished in an aging model. It’s certainly interesting work, highlighting an important mechanism that certainly has relevance in today’s society. Overall the enthusiasm is high for the work with minor thoughts included below.

Minor Thoughts:

Was the White Adipocyte Diameter any different in the 12 month groups?

Response 1: The authors thank so much for the comments from this reviewer. For logistical and financial reasons, it was not possible to evaluate adipocyte diameter at the age of 12 months. Thus, unfortunately. it is not possible to answer this question.

Comments 2: Extra comma’s in first section of page 7. “stimulation under both SD and HFD when compared to WT mice.”

Response 2: Thank you for your observation. We removed the extra commas identified (line 263).

Comments 3: When were the mice placed on the HFD and at what age? It’s unclear if the mice were placed on the HFD at 4ish months of age and then a subset were sacrificed at 6 months of age, with the rest remaining on the diet until they were 12 months. Or if mice were placed on the diet 20-22 weeks before 6 months and 12 months of age and then subsequently sacrificed at those ages. I would just clarify that point in the methods.

Response 3: In accordance with the suggestion, we clarify that point in the methods. The mice were placed on the high-fat diet 20-22 weeks before 6 and 12 months of age and then subsequently euthanized at those ages, as described in lines 133-135 in the methods.

Comments 4: Is the 90 minute data point for Figure 2B missing? It is mentioned in methods.

Response 4: Thank you very much for your observation. Our apologies for the mistake. The 90 minutes of lipolysis in vitro was not evaluated and, therefore, is not shown in Figure 2B. We took it out (line 177). We also corrected the Insulin sensitivity test times in the methods: we removed 90 and 120 minutes, as well as added 45 minutes (line 150), according to figure 3C-D.

Comments 5: Table 2 has the columns too close together and it’s quite challenging to read. Consider either highlighting different columns or provide greater separation.

Response 5: Thank you for your observation. The page orientation of table 2 in the Word file is in "Landscape" format, due to the large amount of information. It is possible that in the Journal edition, the page has changed to "Portrait" format and the table has been modified. Therefore, we will ask the Journal to keep the "Landscape" format so that the configuration of Table 2 does not change.

Comments 6: There is some data missing from the 12 months groups, namely Figure 4 and food intake/water and total energy expenditure in Table 2. Were these variables not measured or what was the rational in leaving these out?

Response 6: For logistical and financial reasons, it was not possible to evaluate all parameter in samples at the age of 12 months, unfortunately.

Comments 7: Consider the fact that both ACE2 deficiency and obesity have been characterized as an accelerated aging phenotype (PMID:3547169, 23160880). When placed together, this could explain why some of the beneficial effects of ACE2 KO disappear in the 12 month age group. These concepts may warrant being included in the discussion and provide a rational for the differences in the age groups.

Response 7: The authors are grateful for the important information that helps to explain and justify our results. Thus, we have added it to the discussion (lines 459-462), as well as added another reference (lines 719-721).

Round 2

Reviewer 1 Report

Comments and Suggestions for Authors

The quality of manuscript has improved.